



# Validation of 10-year SAO OMI Ozone Profile (PROFOZ) Product Using Aura MLS Measurements

Guanyu Huang[1], Xiong Liu[1], Kelly Chance[1], Kai Yang[2], Zhaonan Cai[1]

1. Harvard-Smithsonian Center for Astrophysics, Cambridge, MA, USA
2. Department of Atmospheric and Oceanic Science, University of Maryland, College Park, Maryland, USA

*Correspondence to:* Guanyu Huang (guanyu.huang@cfa.harvard.edu)

**Abstract.** We validate the Ozone Monitoring Instrument (OMI) ozone profile (PROFOZ) product including ozone profiles between 0.22-261 hPa and Stratospheric Ozone Columns (SOCs) down to 100, 215, and 261 hPa from October 2004 through December 2014 retrieved by the Smithsonian Astrophysical Observatory (SAO) algorithm

against the latest Microwave Limb Sound (MLS) v4.2x data. We also evaluate the effects of OMI row anomaly (RA) on the retrieval by dividing the data set into before and after the occurrence of serious RA, i.e., pre-RA (2004-2008) and post-RA (2009-2014). During the pre-RA period, OMI ozone profiles agree very well with MLS data. The global mean biases (MBs) are within 3% between 0.22-100 hPa and negative 3-9% for lower layers, and the standard deviations (SDs) are 3.5-5% from 1-40 hPa, 6-10% for upper layers and 5-20% for lower layers, after

applying OMI averaging kernels to MLS data. OMI shows latitude and solar zenith angle (SZA) dependent biases, but MBs and SDs are mostly within 10% except for low/high altitudes of high latitudes/SZAs. During the post-RA period, OMI retrievals degrade slightly between 5-261 hPa with MBs and SDs typically larger by 2-5%, and degrade much more, with larger MBs by up to 8% and SDs by up to 15% for pressure less than ~5 hPa, where the MBs are larger by 10-15% south of 40°N due to the blockage effect of RA and smaller by 15-20% north of 40°N due to the

solar contamination effect of RA. The much worse comparison at high altitudes indicates the UV-1 channel of pixels that are not flagged as RA is still affected by the RA. During the pre-RA period, OMI SOCs show very good agreement with MLS data with global mean MBs within 0.6% and SDs of 1.9% for SOCs down to 215 and 261 hPa and of 2.30% for SOC down to 100 hPa. Despite clearly worse ozone profile comparison during the post-RA period, OMI SOCs only slightly degrade, with SDs larger by 0.4-0.6% mostly due to looser spatial coincidence criterion as

a result of missing data from RA and MBs larger by 0.4-0.7%. The retrieval comparison indicates significant bias trends, especially during the post-RA period. The spatiotemporal variation of the retrieval performance suggests the need to improve OMI's radiometric calibration to maintain the long-term stability and spatial consistency of the PROFOZ product. The good comparison with SOC down to 261 hPa supports that MLS ozone at 261 hPa, recommended for further evaluation by the MLS team, is suitable for scientific use.





## 1   Introduction

The Dutch-Finnish built Ozone Monitoring Instrument (OMI) on board the NASA Earth Observing System (EOS) Aura satellite has been making useful measurements of trace gases including ozone and aerosols since October 2004. As for ozone, there are two independent operational total ozone algorithms (Bhartia and Wellemeyer, 2002; Veefkind et al., 2006) and two ozone profile algorithms. Of the two ozone profile algorithms, one is the operational algorithm developed at KNMI (van Oss et al., 2001) producing the OMO3PR product, and the other one is a research algorithm developed at Smithsonian Astrophysical Observatory (SAO) by Liu et al. (2010b) producing the PROFOZ product. Both KNMI and SAO algorithms use the optimal estimation technique to retrieve ozone profiles from the spectral range 270-330 nm, but they have significantly different implementation details (e.g. radiometric calibration, radiative transfer model simulation, a priori constraint, retrieval grids, and additional retrieval parameters) (Liu et al., 2010b). The PROFOZ product has been produced in the OMI operational Science Investigator-led Processing System (SIPS) for the entire OMI data record. This product is publicly available at the Aura Validation Data Center (AVDC) (http://avdc.gsfc.nasa.gov/index.php?site=2045907950). In the prequel of this paper, we performed a comprehensive and global assessment of the long-term quality of PROFOZ product (surface to ~ 7 hPa) using ozonesonde observations and showed good agreement with sondes especially in the tropics and mid-latitudes, with mean biases less than 6% and standard deviations of 5-10% for pressure less than 50 hPa and up to 18 (27%) for altitudes below in the tropics (mid-latitudes), despite time-dependent biases especially after the occurrence of serious OMI Row Anomaly (RA) in January 2009 and some biases depending on latitude, season, solar zenith angle (SZA) and cross-track positions (Huang et al., 2017). In this study, we complement the ozonesonde validation of this product with stratospheric ozone data measured by Microwave Limb Sounder (MLS) also aboard the Aura satellite.

Although we used many ozonesondes globally in our previous OMI validation study, there were a few limitations. First, the number of ozonesonde observations is limited and their geographical and temporal samplings are uneven. Only ~10,500 out of ~27,000 ozonesonde profiles are eventually used after OMI/ozonesonde data screening and cloud filtering. Most of them are in the northern mid-latitudes and the tropics, with much fewer observations in the southern middle and high latitudes. Second, the accuracy of ozonesonde observations depends on data processing technique, sensor solution, and instrument type, and the availability of correction factors (Smit et al., 2007; Thompson et al., 2007). Consequently, station-to-station biases may add uncertainties to the OMI ozone validation results (Huang et al., 2017; Worden et al., 2007). Third, ozonesonde measures only from the surface up to ~7 hPa and ozonesondes' burst pressures can vary from sonde to sonde. We used sondes that burst at 7-12 hPa and validated Stratospheric Ozone Columns (SOCs) integrated from tropopause to the OMI layer below sonde burst altitude. Therefore, the top part of OMI ozone profile was not evaluated, but also SOCs were not entirely validated due to the missing ozone information above burst altitudes, which on average consists of ~14% of SOC, ranging from ~6-33% (Huang et al., 2017). This is also why our ozonesonde validation paper focused on ozone evaluation in the troposphere. Furthermore, it has been suggested that OMI measurements at shorter wavelengths in the UV-1 channel may have been affected by the RA at all cross-track positions including those not flagged as RA pixels





(Personal communication with S. Marchenko). As the shorter part of radiance spectra in the UV-1 channel mainly contributes to ozone retrieval in the middle and upper stratosphere, it is necessary to evaluate ozone at this altitude range to understand the RA impacts on the UV-1 channel. Therefore, using ozonesondes only cannot fully validate the retrieval quality of the PROFOZ product, understand the impacts of RA, and assess the radiometric calibration over the entire UV channel.

MLS has been measuring stratospheric ozone since its launch in 2004. This stratospheric ozone product has been shown to have high accuracy and long-term stability by using multiple observations (Froidevaux et al., 2008; Hubert et al., 2016; Jiang et al., 2007). MLS v2.2 and v3.3 ozone data down to 215 hPa have been used to evaluate earlier versions of our OMI ozone profile retrievals as well as SOCs down to 100 hPa and 215 hPa (Bak et al., 2013; Bak et al., 2016; Liu et al., 2010a) for limited time periods (1 year or less) and were demonstrated to be an excellent source to validate OMI stratospheric ozone profiles due to MLS's close collocation with OMI, finer vertical resolution and high quality. In this study, we applied the same methodology to validate stratospheric ozone profiles and SOCs of 10-year PROFOZ product with the recently released MLS v4.2x data, quantifying OMI stratospheric ozone biases spatiotemporally, assessing its long-term performance, and the impacts of RA on the retrievals. In this new version, the pressure range of useful data has been extended from 215 hPa down to 261 hPa although the ozone at 261 hPa is still marked as "requires further evaluation" (Livesey et al., 2015). As MLS stratospheric ozone profiles and SOCs have been used to derive tropospheric ozone column (TOCs) using the Tropospheric Ozone Residual (TOR) method (Jing et al., 2006; Schoeberl et al., 2007; Yang et al., 2007; Ziemke et al., 2006; Ziemke et al., 2014), this extension of useful data range to lower altitudes of the critical Upper Troposphere and Lower Stratosphere (UTLS) region has potential to improve the derived TOCs using the various OMI/MLS TOR approaches. Therefore, we extend the ozone profile comparison down to 261 hPa and compare SOC down to 261 hPa from top of the atmosphere. Such comparison will provide a quantitative evaluation of the quality of MLS data at this lower level relative to those at higher levels (e.g., 215 hPa). This paper is organized as follows: Section 2 describes OMI and MLS data; the validation methodology is introduced in Sect. 3; Section 4 presents results, analysis, and discussions; Section 5 concludes this study.

## 2 Data

### 2.1 OMI and OMI Ozone Profile Retrievals

OMI is a nadir-viewing pushbroom UV/visible spectrometer aboard the NASA EOS Aura satellite that was launched into a sun-synchronous orbit in July 2004. It measures backscattered radiances in three channels covering the 270-500 nm wavelength range (UV-1: 270-310 nm, UV-2: 310-365 nm, visible: 350-500 nm) at spectral resolutions of 0.42-0.63 nm (Levelt et al., 2006) with daily global coverage. There are 60 cross track-positions for UV2 and visible channels, and 30 cross-track positions for UV1 channel due to the weaker signals. The nadir spatial resolution of 13 km × 24 km (along × across track) for UV2 and visible channels, and 13 km × 48 km for UV1 channels. Our PROFOZ product is processed by coadding 4 UV-1 pixels along the track, resulting in product nadir





spatial resolution of 52 km × 48 km. More details on the PROFOZ product and ozone profile retrieval algorithm can be found in Huang et al. (2017) and Liu et al. (2010b).

As aforementioned, certain cross-track positions in OMI data have been affected by the RA since June 2007 (Kroon et al., 2011). The RA has spread to other rows and shifted with time since January 2009, and it became more serious that some of these positions are not recommended for scientific studies. Figure 1 shows the monthly mean number of OMI ozone profile retrievals relative to that in January 2005 as a function of UV-1 cross track position and time (from October 2004 to December 2014), indicating the spatial and temporal distribution of RA affected pixels. The OMI pixels between 13 and 21 cross-track positions are mostly not processed due to RA flagging after 2009. Positions 23-27 were partially (a portion of an orbit) affected by RA except that positions 25-27 were mostly affected by RA during 2009-2011 and positions 22-23 were mostly affected by RA during summer and fall 2011. The effects of RA on the lack of retrievals at certain positions are significant to our OMI/MLS coincidences and will be further discussed in Sect. 3.

To screen out OMI profiles for validation, we applied the same criteria for the retrieval fitting quality as Huang et al. (2017) i.e., applying time-dependent thresholds and selecting retrievals with Root Mean Squire (RMS) of the fitting residuals smaller than the sum of monthly mean RMS and its $2\sigma$. However, the other criteria regarding cloudiness, SZA, and cross-track position are different from Huang et al. (2017). The threshold of effective cloud fraction is removed due to its limited impacts on stratospheric ozone retrievals (Liu et al., 2010a). The SZA threshold is changed from less than 75° to less than 88°, same as the SZA criterion used by Liu et al. (2010a). At SZA larger than 75°, although the retrieval sensitivity to tropospheric ozone is significantly reduced to almost 0, reduction in stratospheric ozone retrieval sensitivity due to reduced signal is offset by increased vertical sensitivity to stratospheric ozone as a result of longer path length in the stratosphere. The cross-track position is also removed because MLS collocates with OMI at an almost fixed cross-track position although the position varies with latitude.

## 2.2 MLS and MLS ozone retrievals

MLS, co-located with OMI aboard the NASA EOS-AURA satellite, is a forward-looking microwave limb sounder that measures thermal emission at millimeter and sub-millimeter wavelengths to observe vertical profiles of atmospheric trace gases, temperature, pressure, and other constituents. It takes measurements along-track, and performs 240 limb scans per orbit, providing ~3500 profiles daily during both daytime and nighttime (Waters et al., 2006). Measurements are taken 7 min ahead of OMI for the same locations during daytime orbital tracks.

The MLS v2.2 products have been validated to be highly accurate using multiple correlative measurements and have been widely used in many studies (Froidevaux et al., 2008; Jiang et al., 2007; Livesey et al., 2008). In MLS v3.3/3.4 versions, stratospheric ozone profiles are generally very similar except that the profile is reported on a finer vertical grid and the bottom pressure level with scientifically useful value increases from 215 hPa to 261 hPa. Ozone in the upper troposphere shows smaller biases under clear-sky conditions but more vertical oscillations under thick cloudy conditions. The vertical resolution increases from ~3 km to 2.5-3 km at most stratospheric altitudes. The across-track resolution is kept ~6 km, but the along-track resolution has been updated from ~200 km in v2.2 to 300-450 km in v3.3/3.4 versions, depending on altitude (Livesey et al., 2015). The latest MLS version 4.2x ozone





product, released in February 2015, is used in this paper for the validation of our PROFOZ product. MLS v4.2x ozone profiles are also generally very similar to previous versions. One of the major improvements of MLS v4.2x is the handling of contamination from cloud signals in trace gas retrievals that results in significant reduction in the number of spurious MLS profiles in cloudy regions and a more user-friendly and efficient screening of cloud-contaminated measurements. Furthermore, the MLS ozone products have been improved through additional retrieval phases and reduction in interferences from other species (Livesey et al., 2015).

The recommended altitude range of MLS v4.2x data is 0.02–261 hPa, but ozone at the 261 hPa level still requires further evaluation. We use MLS 0.22–261 hPa to compare with our OMI PROFOZ product. The comparison at 261 hPa serves as a cross-evaluation of both OMI and MLS data. According to the data screening criteria of the MLS v4.2 data document (Livesey et al., 2015), we use only profiles with even values of the "Status" field, "Quality" fields greater than 1.0 and "Convergence" fields less than 1.03. The vertical and horizontal resolutions are almost the same as those in v3.3/3.4, mentioned above in this section. The precision is estimated to be ~7-30 % (0.1 - 0.2 ppmv) at 0.2 -1 hPa, 2-3% (0.06 -0.15 ppmv) at 2-46 hPa, and 0.02-0.04 ppmv (3-100% due to the high ozone variation) at 68 – 215 hPa. The accuracy is estimated to be 0.05-0.3 ppmv (~3-10%) in most of the stratosphere (0.2-68 hPa), approximately 0.02-0.05 ppmv + 5-20% in the 100-215 hPa region. The precision of the ozone column (for a single profile) down to 100-215 hPa is approximately 2% or less and the accuracy is estimated to be 4% (Livesey et al., 2015).

## 3    Methodology

The ideal OMI-MLS coincident criterion is that the center of an MLS footprint lies within a collocated OMI footprint, same as in Liu et al. (2010a). Then the spatial difference arises from unequal horizontal resolutions: 52 km × 48 km for OMI vs. 300-500 km × 6 km for MLS, and the time difference is 7 minutes. The OMI cross-track position collocated with MLS data varies with latitude, ranging from UV-1 position 20 in the tropics to position 15 at high latitudes (Liu et al., 2010a). Before the occurrence of serious RA (2004-2008, pre-RA period), we can always find the ideal OMI-MLS collocation. However, after the serious occurrence of RA (2009-2014, post-RA period), retrievals for some of these positions are not available as shown in Fig. 1. Consequently, we must downgrade our OMI-MLS coincident criterion to nearest OMI footprint from the center of an MLS footprint. In the tropics, the collocated OMI position changes to position 14 during 2009-2010 and to position 12 after 2010; at high latitudes, the collocated OMI position changes to position 22 after 2009 except for position 24 during summer and fall 2011. These different OMI-MLS coincidence criteria between pre-RA and post-RA periods, and the cross-track dependent retrieval biases as shown in Huang et al. (2017) might influence our evaluation of the OMI long-term stability. Similar to Huang et al. (2017), we will conduct the comparison for both pre-RA and post-RA periods, respectively to evaluate the impacts of RA on the retrievals. To find out whether our validation is affected by different coincident criteria, we will also apply the same post-RA coincidence criterion (position 12 in the tropics and 22 at high latitudes) to pre-RA measurements by masking out positions 13-21 (i.e., pre-PA period with post-RA mask). The comparison differences between using ideal coincidence and post-RA coincidence for the pre-RA period will provide the impacts of coincidence criteria on the comparison.




Although MLS v4.2x ozone data in the vertical range 0.02-261 hPa is recommended for scientific use, the top layer in our OMI retrievals is a broad layer from 0.35 hPa to top of the atmosphere. We mainly use MLS data to validate our retrievals from 0.22 to 261 hPa for avoiding the large interpolation errors in this broad layer.

MLS and OMI ozone profiles have different vertical grids and resolutions; we follow the approach used in Liu et al. (2010a) to account for these vertical differences. MLS ozone profiles in volume mixing ratios are integrated to partial ozone columns using a procedure provided by the MLS team. To compare OMI retrievals with original MLS data directly, OMI partial ozone columns are integrated to the MLS vertical grids. To account for the different resolutions, MLS partial ozone columns are first interpolated to OMI vertical grids, and then degraded to the OMI vertical resolution by assuming MLS data to be the truth ($X_{MLS}$), and simulating the expected retrieval ($X'_{MLS}$) from our OMI algorithm using OMI averaging kernels (AKs):

$$X'_{MLS} = X_a + A(X_{MLS} - X_a) \tag{1}$$

where $Xa$ is the a priori ozone profile used in OMI retrievals and $A$ is the AK matrix. The differences between $X'_{MLS}$ and $X_{MLS}$ are the estimated OMI smoothing errors with relative to MLS data, although we note that errors in MLS data could affect the estimates. The convolved MLS data are then interpolated back to the original MLS grids and compared with OMI retrievals interpolated to the MLS grids. In addition to the comparisons between OMI and convolved MLS data, we also compare OMI with original MLS data. The comparison with original MLS data indicates how well our retrievals can represent actual ozone data. The comparison with convolved MLS data removes the OMI smoothing error-related component of the differences between MLS profiles and OMI a priori, which is significant in the UT/LS region, and therefore allows us to better identify other sources of OMI/MLS errors. The normalized difference is defined as (OMI-MLS)/OMI a priori × 100%. OMI a priori is used in calculation instead of MLS values, because the relative differences with respect to MLS data could be unrealistically large due to the small MLS values in the tropical upper troposphere and lower stratosphere (Liu et al., 2010a).

For validation of OMI SOCs, original/convolved MLS SOCs are integrated from original/convolved MLS ozone profiles between 0.22 and corresponding pressure levels (100 hPa, 215 hPa or 261 hPa). Ozone column from 0.22 hPa to the top of the atmosphere, which is included in the OMI SOCs, is generally less than 0.1-0.2 DU, and therefore negligible (Liu et al., 2010a). We compare OMI SOCs with both original and convolved MLS SOCs although most of the SOC comparison results are for comparison with original MLS data as the smoothing errors in SOCs are relatively small.

Although we have applied RA flagging in the OMI level 1b data, we notice unusually large retrieval anomalies during several time periods. Figure 2 shows the deseasonalized time series of monthly mean OMI and MLS SOC215 in 60°N-30°N, 30°N -30°S and 30°S-60°S, respectively. It clearly shows large positive OMI retrieval anomalies of up to 20 DU (10 DU) in the tropics and southern mid-latitudes during July - October 2011 (July - December 2014), and smaller but still noticeable positive anomalies at all latitudes during March - October 2009. The reason for such sporadic large retrieval anomalies in cross-track positions not flagged as RA pixels is still not clear, but likely associated with the impacts of RA. And it is recommended not to use the data over these periods. Consequently, we exclude the use of OMI PROFOZ products during these three time periods in this validation. However, these anomalies are not readily detected in our OMI/ozonesonde comparison (Huang et al., 2017). This is probably



because the number of OMI-ozonesonde pairs is much smaller than that of OMI-MLS pairs during these time periods, and the OMI/sonde SOCs are integrated from tropopause up to ~7 hPa only, missing to include ozone at pressure less than 7 hPa.

For both profile and SOC comparisons, we will first show global comparison during pre-RA and post-RA periods, respectively. The pre-RA comparison will be done using both ideal collocation criterion and post-RA mask. Then the comparisons will be done as a function of latitude, and SZAs in the southern and northern hemispheres, in which results with the ideal coincidence criterion is applied for the pre-RA period.

To evaluate the long-term performance of our ozone profile retrievals, we analyze the monthly mean biases (MBs) of the OMI/MLS differences as a function of time for five different latitude bands: northern high latitudes (90°N-60°N), northern mid-latitudes (60°N-30°N), tropics (30°N-30°S), southern mid-latitudes (30°S-60°S), and southern high latitudes (60°S-90°S). We then derive linear regression trends for the entire period (2004-2014), the pre-RA (2004-2008), and post-RA (2009-2014) periods, respectively. In this evaluation of long-term performance, the pre-RA comparison is done using the post-RA mask, consistent with the comparison during the post-RA period.

## 4    Results and Discussions

### 4.1    Comparison of OMI and MLS Ozone Profiles

Figure 3 (a) and (b) shows OMI/MLS ozone profile comparison over the globe during the pre-RA period (2004-2008) with ideal coincidence criterion. When compared with the original MLS profiles (blue lines), OMI agrees well with MLS, with global mean biases (MBs) within 6% from 1-150 hPa, positive biases of up to 12% for upper layers and negative biases of up to 25% for the bottom layers. The corresponding standard deviations (SDs) are within 4-7% from 1-50 hPa, increasing to ~13% for the top layers and to 27%-42% between 100-261 hPa. These results exhibit significant improvements over the OMI a priori as shown in black with 1-5% smaller SDs from ~1-261 hPa, while the larger SDs from pressure less than 1 hPa than those of OMI a priori/MLS comparison indicate that the combined errors from OMI and MLS exceed the natural variability of ozone.

The smoothing errors (green lines), estimated by assuming MLS data as the truth, are generally within 3% from 0.5-215 hPa, increasing to -12% at the 215-261 hPa layer and to 12% at pressure less than 0.5 hPa. Their SDs are 2-10% from 0.2-70 hPa, increasing up to 20% and 35% for the bottom two layers. The smoothing errors dominate OMI/MLS variances over the pressure range of 5-261 hPa and the larger OMI/MLS variance at pressure less than 1 hPa are likely due to the large combined OMI/MLS errors. The smoothing errors are generally consistent with OMI retrieval estimates of smoothing errors except for pressure above 50 hPa, where MLS derived smoothing errors are significantly larger. Differences due to different spatiotemporal footprints and OMI/MLS systematic errors may contribute to larger MLS-derived smoothing errors.

After removing smoothing errors through convolving MLS profiles with OMI AKs (red lines), OMI retrievals show good agreement with MLS data (red lines) to within 3% from 0.22-100 hPa. For pressure greater than 100 hPa, OMI has negative biases of 3-9% compared to MLS data. These negative biases are not entirely from OMI because MLS ozone has been found to have positive biases in this altitude range relative to other correlative measurements





(Froidevaux et al., 2008; Liu et al., 2010a). The SDs are 3.5-5% from 1.5-40 hPa, 6-10% for upper layers and 5-20% for lower layers.

The comparison is generally similar to the OMI/MLS comparison in 2006 in Liu et al. (2010a) although both OMI and MLS versions are different and the comparison in this study is done at a finer vertical grid (due to the finer

grid in MLS v4.2x data than in MLS v2 data). The comparison in the bottom layer (215-261 hPa) has larger negative MBs and larger SDs compared to that in the layer above (150-215 hPa). This is probably due to a combination of larger retrieval errors in both OMI and MLS data and larger ozone variability.

Figure 3 (c) and (d) show similar comparison during the pre-RA period but with post-RA mask coincidence criterion, with the OMI/MLS comparison in Figs. 3 (a) and (b) also shown as dashed lines. The results are very

similar, larger by less than 0.5% in both MBs and SDs. This indicates that the choice of OMI-MLS coincident criteria has negligible effects on the ozone profile comparison.

Figure 3 (e) and (f) show similar comparison during the post-RA period (2009-2014). The OMI retrievals clearly exhibit degrading data quality indicated by larger MBs and SDs than those during the pre-RA period. The largest changes occur in the upper layers at pressure less than ~5 hPa; MBs increase by 2-5% and SDs increase by

up to ~15%. Such worse comparison at higher altitudes, where the retrieval sensitivity mainly comes from shorter wavelengths (< 300 nm) in the UV-1 channel, indicates that RA significantly affects the UV-1 channel, although those OMI pixels are not flagged as RA pixels. At lower altitudes, OMI still shows good comparison with MLS data although the MBs and SDs can be larger by 2-5%.

Figure 4 shows the MBs and SDs of the differences between OMI and MLS (convolved with OMI AKs) ozone

profiles as functions of latitude during the pre-RA (with ideal coincidence) and post-RA periods, respectively. During the pre-RA period, the MBs are generally within 10% except for positive biases of up to ~20% from 40-80 hPa, and negative biases of up to ~20% from below at high latitudes, and negative biases of up to ~15% in the upper stratosphere of northern high latitudes. Clearly, the MBs show large oscillations at high latitudes likely due to both larger retrieval errors and ozone variability. In addition, OMI still shows some systematic cross-track position

dependent biases. As the OMI/MLS coincident position varies with latitude, cross-track dependent biases in our OMI retrievals can be up to 5% in the UTLS and within a few percent at higher altitudes. These errors will also contribute to the latitude dependence of OMI/MLS comparison. The patterns of SDs are similar to that in Figure 3 (b), but typically with larger values at high latitudes. In 1-40 hPa, the SDs increase from 2-4% in the tropics to 4-8% at high latitudes. At pressure less than 1 hPa or greater than 50 hPa, the SDs increase from 5-10% in the tropics to

~30% at high latitudes. The patterns in MBs and SDs are quite symmetric between two hemispheres. During the post-RA period, the most significant changes in MBs occur at pressure less than ~5 hPa, where MBs are larger by 10-15% south of 40°N and smaller by 15-20% north of 40°N so that the patterns, especially in MBs, are not symmetric anymore. This supports that the RA has different effects during different portions of the orbits. According to the analysis of RA behavior in Schenkeveld et al. (2017), in the northern part of the orbits, the effect is

dominated by the solar contamination that increases radiance values and thus causes negative ozone biases; in the rest of the orbit, the effect is dominated by the blockage effect that reduces radiance values and thus causes positive





ozone biases. The SDs during the post-RA period are typically larger by 2-5% but by >10% at high altitudes than those during the pre-RA period.

Figure 5 shows comparisons similar to those in Figure 4 except as a function of SZA in southern and northern hemispheres, respectively. During the pre-RA period, OMI has good agreement with MLS in 1-100 hPa for SZA less than 80°, with MBs of <10% and SDs of 3-15% in both hemispheres, but with larger MBs and SDs at larger SZAs. As SZA correlates with latitude, OMI cross-track dependency will contribute to the SZA dependence of the comparison. The patterns in MBs and SDs are again quite symmetric between the two hemispheres despite different ozone fields except that the MBs are more negative in the northern hemisphere. The symmetry suggests that these biases are likely caused by SZA-dependent errors such as errors due to OMI staylight errors and radiative transfer calculations errors. During the post-RA period, in addition to the larger MBs and SDs as shown in Figure 4, the patterns especially in the MBs are not symmetric anymore as a result of the blockage effect of the RA in the southern hemisphere, and lower latitude (smaller SZA) of the northern hemisphere and the solar contamination effect of the RA in the higher latitude (larger SZA) of the northern hemisphere.

## 4.2 Comparisons of OMI and MLS Stratospheric Ozone Columns (SOCs)

Figure 6 shows a scatter density plot of OMI and MLS SOCs (SOC100, SOC215, SOC261) without applying OMI AKs during the pre-RA (2004-2008 with ideal coincidence criterion and post-RA mask) and post-RA periods, respectively. The corresponding comparison statistics are shown in black legends. All OMI SOCs show excellent agreement with MLS data even during the post-RA period. The correlation coefficients are typically within 0.96-0.99 and the linear regression slopes are within 0.04 to 1. MBs are typically within 1.6 DU (0.7%) except for 3.1 DU (1.3%) in SOC100 during the post-RA period, and SDs are within 7.0 DU (2.71%).

OMI SOCs show the best comparison with MLS SOCs during 2004-2008 with ideal coincidence as clearly seen by the smallest spread of the scatters, the smallest SDs (5.3-5.6 DU or 1.9-2.3%) and the highest correlation coefficients (0.98-0.99). The MBs are within 1.3 DU (0.6%). With the lower boundary from 100 hPa to 215 hPa to 261 hPa, MBs decrease from 1.3 DU to 0.4 DU to -0.2 DU due to negative biases in the bottom three layers as shown in Fig. 3, SDs decrease from 5.6 (2.3%) to 5.4 DU (1.9%) to 5.3 DU (1.9%). The SOC215 comparison is significantly better than that during 2006 shown in Liu et al. (2010a), where the MBs are -1.8±7.7 DU. This might reflect the improvement of both OMI and MLS over previous versions. Using the post-RA mask for the pre-RA period, the comparison only slightly degrades, with MBs larger by up to 0.3 DU, SDs larger by 0.9-1.7 DU (0.4-0.6%) and correlation smaller by 0.01. During the post-RA period, the MBs systematically increase by 1-1.4 DU over those during the pre-RA period with post-RA mask, but the SDs and correlation do not change much. Despite of the degradation in ozone profile comparison especially at pressures less than ~5 hPa during the post-RA period, the overall SOC comparisons do not degrade much except for the systematic increase of MBs by ~1 DU. Also, the comparison for SOC261 is very similar to that for the SOC215, indicating that ozone in 215-261 hPa and SOC down to 261 hPa can be used for scientific use, e.g., to derive tropospheric ozone columns from OMI/MLS using the TOR method.





The comparisons with AKs applied to MLS data are not shown but the comparison statistics are shown in red legends in Figure 6. As expected, applying OMI AKs improves the comparison with smaller SDs and better correlation. But the reduction in SDs is very small, typically within 0.3 DU (0.1%), and the increase in correlation is within 0.001. The values of MBs are smaller, typically by less than 0.2 DU for SOC100 and SOC215, and by 0.7

DU for SOC261.

Figure 7 shows the SOC comparisons as a function of latitudes. Generally, SOC100, SOC215, and SOC261 comparison results have similar patterns: small MBs (within 1%) and SDs (within 2%) in the lower latitudes, and some latitude dependence in higher latitudes. The SOC100 MBs show significant latitudinal dependence, typically increasing with latitude. For example, SOC100 MBs during 2004-2008 are within 0.5% between 50°S-50°N areas

and increase up to 4% with the increase of latitude in both hemispheres. Such latitudinal dependence decreases with increasing pressure of the lower boundary. SDs typically increase with latitude for all three SOCs, from within 2% in the tropics to up to ~4% at higher latitudes, mainly due to a combination of larger OMI retrieval errors and larger ozone variability.  During the post-RA period, the MBs are typically more positive and SDs are always larger especially at higher latitude mostly due to the use of looser coincidence criterion; such contrast between pre-RA and

post-RA periods are consistent with that in Figure 6.

Figure 8 shows similar SOC comparisons as functions of SZAs for both southern and northern hemispheres. During the pre-RA period, the patterns of MBs and SDs are relatively symmetric between the two hemispheres. For SOC100, MBs are within 0.5% and do not show much SZA dependence at SZA less than 45°, then generally increase with SZA to up to 2.5%. For SOC215 and SOC261, MBs are within 0.5% and show little SZA dependence

until SZA > 80°, then MBs dramatically increase with SZA to up to ~4%. The SDs are within 2% gradually increase with SZA when MBs does not change much, and then increase more with the increase of SZAs to up to ~5%. During the post-RA periods, the MBs and SDs generally show more SZA dependence, SDs are typically larger especially at high SZAs, and patterns are not symmetric anymore between the two hemispheres. The patterns in the southern hemisphere are still similar to that during the pre-RA period. In the northern hemisphere, MBs are reduced

by up to 2% at SZAs of 65°-80° due to the solar contamination effect of the RA.

Compared to the results of Liu et al. (2010a), the SOC215 comparison during the pre-RA period is more symmetric between the two hemispheres; the SDs show stronger latitude/SZA dependence, and the MBs typically show less latitude/SZA dependence and are closer to zero. The SOC261 comparisons are very similar to the SOC215 comparisons during both pre-RA and post-RA periods. This implies that MLS ozone at 261 hPa is good for

scientific use and SOC261 can be used to improve the TOC derivation using the OMI/MLS TOR method by covering a broader range of the stratosphere in the middle and high latitudes where the tropopause pressure is ~ 261 hPa or larger.

### 4.3 Evaluation of Long-term Performance

Figure 9 shows the linear trends in the relative OMI/MLS biases during 2004-2014, pre-RA (2004-2008) and

post-RA (2009-2014) periods for the five latitude ranges. It should be noted that the three time periods (March – October 2009, July-October 2011, and July-December 2014) with very large OMI/MLS biases are excluded in our



trend analysis. The solid circles indicate significant trends with P values less than 0.05. We can see significant bias trends at most altitudes during 2009-2014 or 2004-2014 for all latitude ranges. The absolute trends are typically larger during 2009-2014 than during 2004-2014 as the latter includes the pre-RA period when the retrievals are more stable. The trends are stronger at northern middle and high latitudes and become smaller at other latitude ranges,

indicating that the dominant solar contamination effect of the RA occurring in the northern portion of the orbit might have a stronger temporal variation. Most of the significant trends are within 3%/year, but there are trends with absolute values of greater than 5% at high altitudes (pressure less than 3 hPa) of 90°N-30°N and at lower altitudes (pressure greater than 125 hPa) of 90°N-60°N. The OMI retrievals are more stable during the pre-RA period. Significant trends do occur at pressure less than 8 hPa especially in the tropics, but the values are typically less than

1%/year except for the upper stratosphere where there are trends of up to -5%/year at 90°N-30°N.

Figure 10 shows monthly mean OMI/MLS SOC215 biases as a function of time for the five latitude ranges. Significantly large MBs occur in the tropics and both mid-latitude regions during March–October 2009, and in the tropics and southern mid-latitude during July–October 2011 and July–December 2014 as have been shown in Figure 2. As mentioned in Sect. 3, such large biases during the three periods are excluded in the trend analysis. Otherwise,

they can drive the trend calculation. The impacts of RA are clearly seen from the much larger temporal variation in the tropics and mid-latitudes even after excluding the above three periods. During the post-RA periods, there are significant linear trends of 0.65-1.16 DU/year except for the tropics, where the SOC215 MBs show large temporal variation, but the overall trend is small, and except for southern high latitudes, where the linear trend is large, but the P value of 0.14 is less significant. For the entire 2004-2014 period, there are significant linear trends of 0.26-0.40

DU/year in the tropics and mid-latitudes. During the pre-RA period, there are statistically significant trends of 0.21-0.34 DU/year in the tropics and northern mid-latitudes, but the trends are much smaller than those during the post RA periods.

The significant trends of OMI/MLS ozone biases at different layers as well as in SOC suggest that the current ozone profile product is not suitable for trend studies, especially during the post-RA period. The retrieval is more

stable during the pre-RA period. The stronger temporal variation of the retrieval performance during the post-RA period is likely associated with the RA evolution. To maintain the long-term stability of our ozone profile product, we need to perform soft calibration similar to Liu et al. (2010b), especially during the post-RA period if the radiometric calibration improvement cannot be done in the level 0-1b processing. As the effects of the RA vary along the orbit (e.g., dominated by the solar contamination effect during the northern portion of the orbit and the

blockage effect during the rest of the orbit), the empirical correction should be derived as a function of both latitude and time.

## 5   Summary and Conclusion

To complement our validation of the 10-year OMI PROFOZ product using ozonesonde observations in a prequel of this paper, we evaluated this product including ozone profiles from 0.22–261 hPa and Stratospheric

Ozone Columns (SOCs) down to pressure levels 100, 215, and 261 hPa (i.e., SOC100, SOC215, SOC261) using MLS v4.2x product during the same period (October 2004-December 2014). To investigate the impacts of Row




Anomaly (RA) on the retrievals, we contrasted the comparison before and after the occurrence of major OMI RA in January 2009, i.e., 2004–2008 (i.e., pre-RA period) and 2009-2014 (i.e., post-RA period). We applied ideal OMI-MLS coincident criterion (i.e., MLS footprint center lies in the footprint of an OMI footprint) in the pre-RA period and the nearest coincident criterion where retrievals for the ideally collocated OMI pixels are not available due to

the RA impacts. To show the impacts of coincident criteria on the comparison, we also conducted the comparison for the pre-RA period using the post-RA coincidence criterion by masking pre-RA retrievals at cross-track positions 13–21 (i.e., pre-RA with post-RA mask). There are unreasonably large OMI-MLS biases during March – October 2009, July-October 2011 and July-December 2014, which was very likely caused by the RA. Therefore, we excluded OMI data during these periods. To better understand retrieval errors and sensitivity, we compared the

retrieved ozone profiles and a priori profile at individual layers with MLS data before and after being degraded to the OMI vertical resolution with OMI averaging kernels (AKs), and characterized the OMI smoothing errors using MLS data. To show the spatial consistency of the retrievals, we performed the comparison as a function of latitude and solar zenith angle (SZA). Finally, we analyzed the monthly variation of the mean biases (MBs) of ozone profiles and SOC215 to examine the long-term stability of our OMI PROFOZ product.

During the pre-RA period, OMI ozone profiles agree well with the original MLS data. The global MBs are within 6% from 1-150 hPa, and up to 12% for the upper layers and up to -25% for the bottom layers. The standard deviations (SDs) range from 4-7% in 1-50 hPa, up to ~13% for pressure <1 hPa and up to 42% for pressure >50 hPa. The large SDs at lower altitudes are mainly due to OMI smoothing errors. After removing smoothing errors by convolving MLS data with OMI AKs, the MBs are within 3% between 0.22 hPa - 100 hPa, negative 3-9% for lower

layers, and the SDs are within 3.5-5% between 1-40 hPa, 6-10% for upper layers and 5-20% for lower layers. The impact of using different coincident criteria on the ozone profile comparison is negligible; using the post-RA mask increases the SDs/MBs by less than 0.5%. During the post-RA period, OMI retrievals become slightly worse between 5-261 hPa with global MBs and SDs typically larger by 2-5% and are much worse for pressure less than ~5 hPa with larger MBs by up to 8% and SDs by up to 15%. The much worse comparison at higher altitudes indicates

that RA significantly affects the UV-1 channel of the OMI measurements, although they are not flagged as RA pixels.

    OMI ozone profiles show latitude and SZA dependent biases with respect to MLS data. During the pre-RA period, the patterns in MBs and SDs are quite symmetric between the two hemispheres despite different ozone fields, which suggests that these biases were likely caused by SZA-dependent errors such as errors due to OMI

staylight errors and radiative transfer calculations errors. MBs are generally within 10%, but show larger oscillations at high latitudes/SZAs with positive biases of up to 20% from 40-80 hPa and negative biases of up to 20% from below. SDs increase from 2-4% at lower latitudes/SZAs to 4-8% at high latitudes/SZAs in 1-40 hPa, and increases from 5-10% to ~30% at pressure less than 1 hPa or greater than 50 hPa. During the post-RA period, the different effects of RA caused asymmetry in the patterns of MBs and SDs between the two hemispheres at pressure less than

~5 hPa, where the MBs are larger by 10-15% south of 40°N due to the blockage effect of RA and smaller by 15-20% north of 40°N due to the solar contamination effect of RA, and the SDs are larger by 10%. For lower altitudes (5-261 hPa), the MBs and SDs are typically larger by 2-5% than those during the pre-RA period.





All OMI SOCs show very good agreement with original MLS data during both pre-RA and post-RA periods. During the pre-RA period, the global mean MBs are within 0.6%, and the SDs are 1.9% for SOC215 and SOC261 and 2.30% for SOC100. Using the post-RA mask only slightly degrades the pre-RA comparison, with MBs larger by up to 0.3 DU, SDs larger by 0.9-1.7 DU (0.4-0.6%). During the post-RA period, OMI SOCs also slightly degrade,

with MBs larger by 1-1.4 DU (0.4-0.7%) and SDs comparable to those during the pre-RA period with post-RA mask. Applying the OMI AKs to MLS data only slightly improves the comparison due to the small smoothing errors in SOCs, reducing the SDs by less than 0.3 DU (0.1%). Similar to the ozone profiles, OMI SOCs show latitude/SZA dependent biases. During the pre-RA period, MBs (SDs) are within 0.5% (2%) in the lower latitudes/SZAs and can increase up to 4% at high latitudes and up to ~6% at SZAs larger than 85°. The MBs in SOC215 and SOC261 show

less latitudinal dependence and show little SZA dependence until SZA > 80° compared to SOC100. The patterns in MBs and SDs are quite symmetric especially for SOC215 and SOC261 as a function of SZA. During the post-RA period, the MBs and SDs generally show more latitude/SZA dependence. The MBs are typically more positive, and the SDs are larger especially at high latitudes/SZAs. The patterns in MBs and SDs are less symmetric due to the various effects of RA along the orbit (i.e., blockage effect south of ~40°N and solar contamination effect north of

~40°N). The SOC261 comparison is very similar to the SOC215 comparison during both pre-RA and post-RA periods, implying the MLS ozone at 261 hPa and MLS SOC261 are good for scientific use and can be used to improve the tropospheric ozone column derivation using various OMI/MLS tropospheric ozone residual methods.

Finally, we evaluated long-term data stability of ozone profile and SOC215 with respect to MLS data and calculate linear regression trends of monthly MBs in five different latitude ranges during pre-RA, post-RA and

2004-2014 periods, respectively. This validation shows significant bias trends or more temporal variation during the post-RA period; significant trends also occur during the pre-RA period, for example for ozone at pressure less than 8 hPa in the tropics and SOC215 in the tropics and northern mid-latitudes, but the magnitudes of the trends or the temporal variation are much smaller compared to those during the post-RA period. The spatiotemporal variation of the retrieval performance suggests the need to improve OMI's radiometric calibration as a function of both latitude

and time especially during the post-RA period to maintain the long-term stability and spatial consistency of our ozone profile product.

*Acknowledgements.* This study was supported by the NASA Atmospheric Composition: Aura Science Team (NNX14AF16G) and the Smithsonian Institution. The Dutch-Finnish OMI instrument is part of the NASA EOS Aura satellite payload. The OMI Project is managed by NIVR and KNMI in the Netherlands. We acknowledge the

OMI and MLS science teams for providing OMI and MLS data. The authors also thank Dr. Pawan K. Bhartia, NASA Goddard Flight Center, for his excellent contribution to this paper.





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



**Figures and Figure Captions**

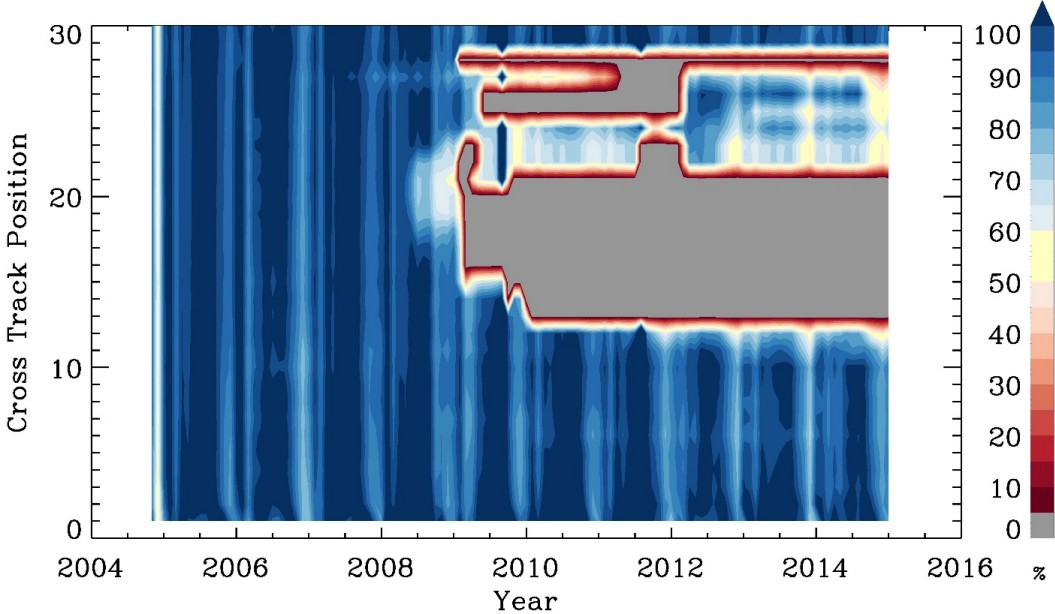

**Figure 1.** The monthly mean number of OMI retrievals at each cross-track position from October 2004 through December 2014, relative to the corresponding monthly mean number of OMI retrievals in January 2005, i.e at the beginning of OMI operation.



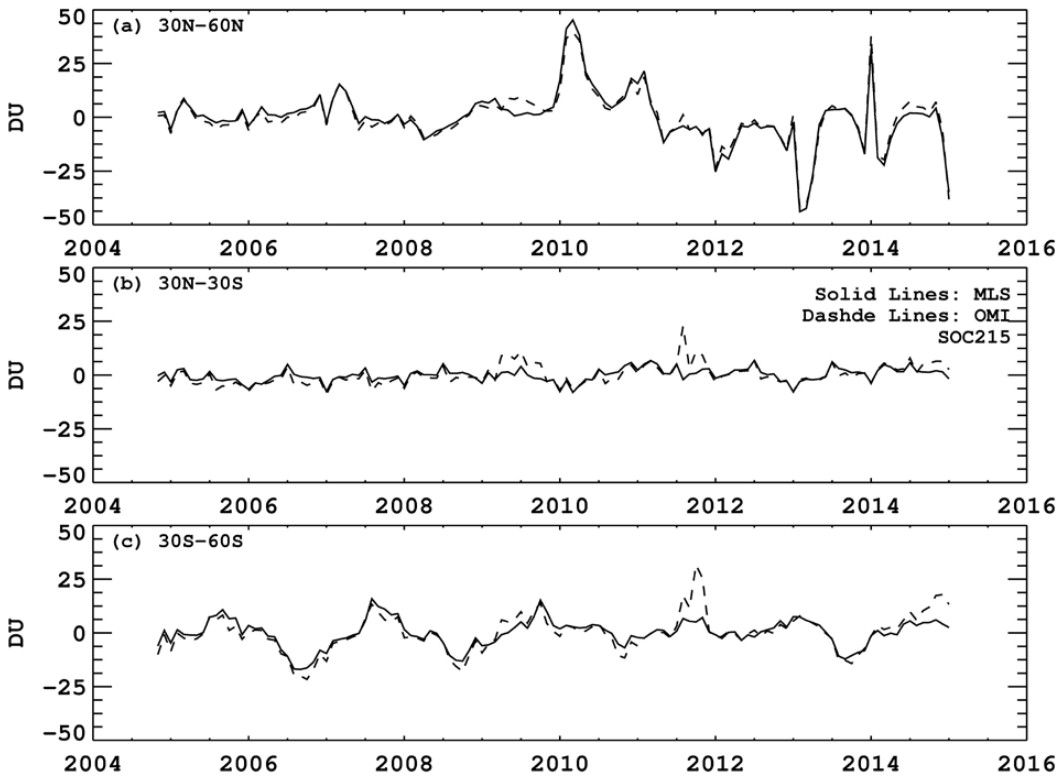

**Figure 2.** Deseasonalized time series of monthly mean MLS (solid lines) and OMI (dashed lines) Stratospheric Ozone Columns down to 215 hPa (SOC215s) in (a) 30°N-60°N, (b) 30°N-30°S and (c) 60°S-30°S, respectively.





**Figure 3.** (left) Global mean biases at each MLS layer and (right) corresponding standard deviations of the differences between OMI and MLS profiles from 0.22 -261 hPa in (top) 2004-2008, (middle) 2004-2008 with post-RA mask and (bottom) 2009-2014, respectively. The black/blue lines compare a priori/OMI retrievals with original MLS profiles. The red lines compare OMI retrievals with MLS profiles after applying OMI Averaging Kernels (AKs). The green lines represent OMI smoothing errors estimated by assuming MLS data as the truth. For contrast, the comparisons of OMI and MLS data based on the ideal collocation criteria in 2004-2008 are plotted in blue and red dashed lines in the middle panels.





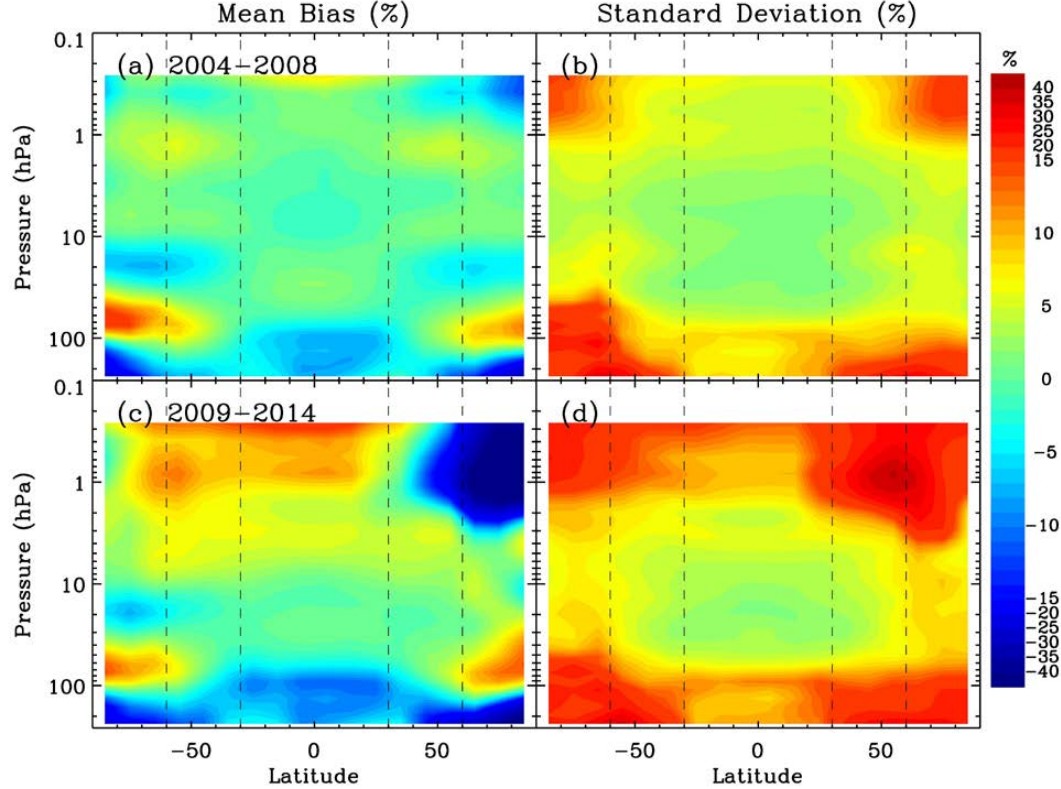

**Figure 4.** (left) Mean OMI/MLS ozone profile biases and (right) corresponding standard deviations as a function of latitudes in (top) 2004-2008 and (bottom) 2009-2014, respectively. OMI averaging kernels are applied to MLS data.





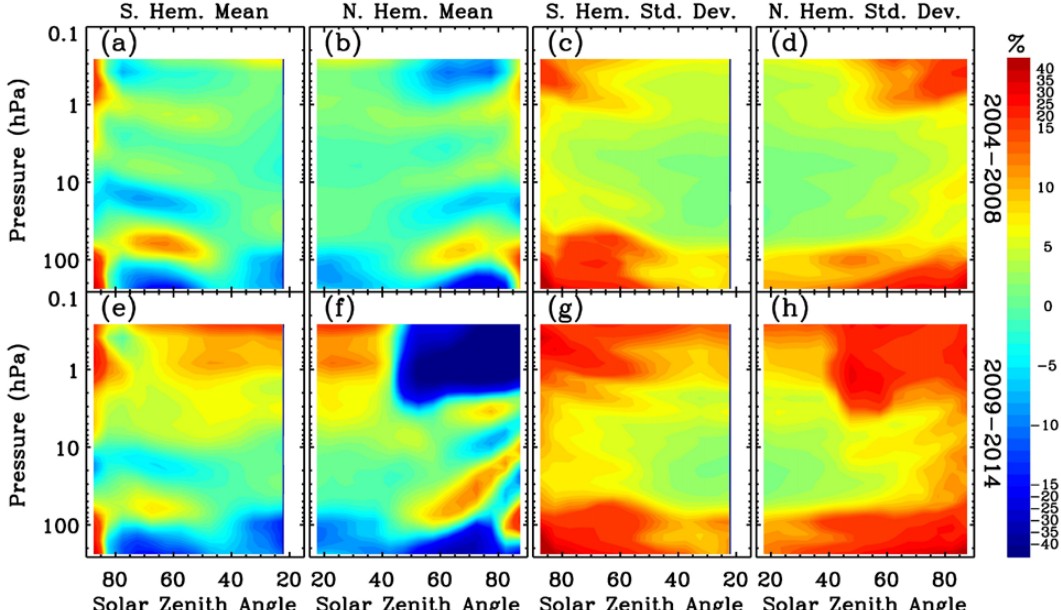

**Figure 5.** Similar to Figure 4, but as a function of solar zenith angles for South and North Hemispheres, respectively.





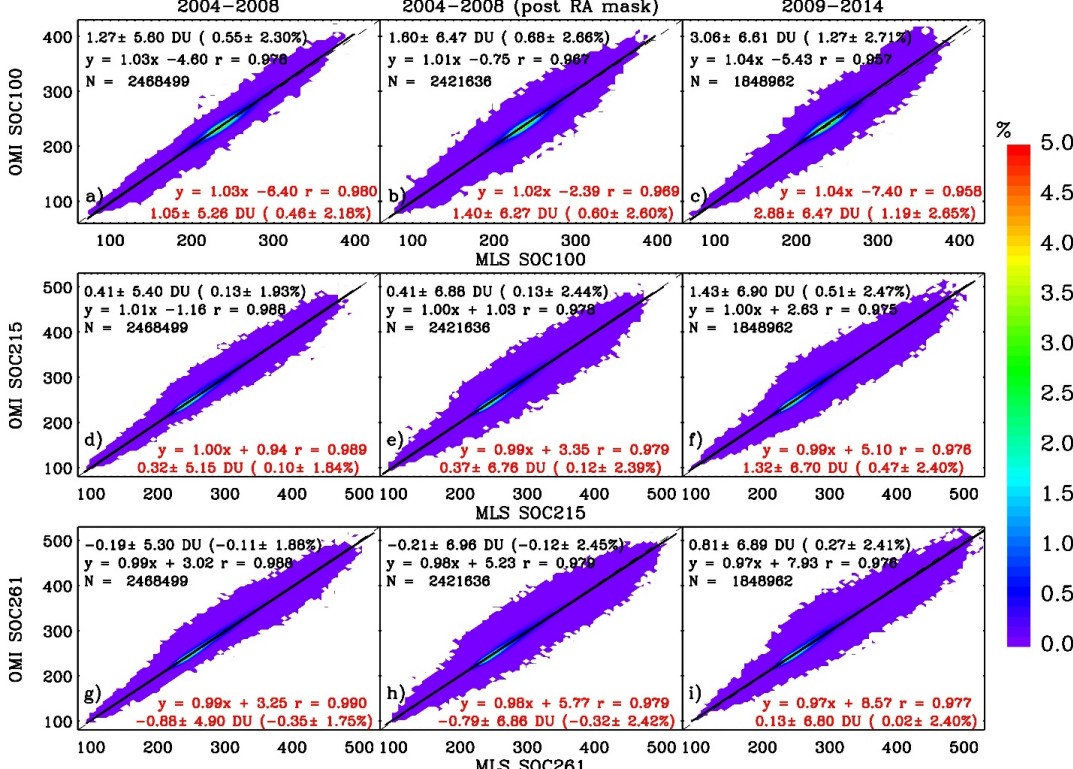

**Figure 6.** Scattering density plots of OMI and MLS stratospheric ozone columns above 100, 215 and 261 hPa (i.e., SOC100, SOC 215 and SOC261), respectively, from the top to bottom panels, during 2004-2008 (left), 2004-2008 with post-RA mask (middle column), and 2009-2014 (right). OMI averaging kernels (AKs) are not applied to MLS data. The comparison statistics of OMI-MLS SOCs are shown in black including mean biases, standard deviations, relative mean biases and standard deviations, correlation coefficients, linear regressions and the number of OMI-MLS pairs. Comparison statistics with OMI AKs applied to MLS data are also provided in red.





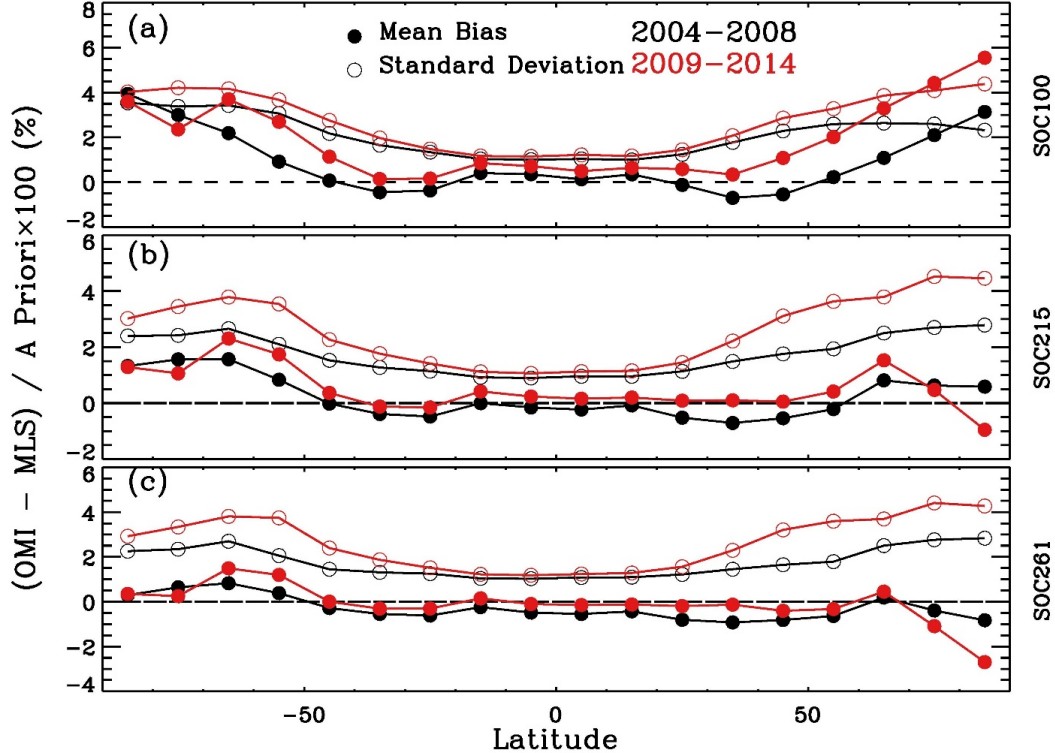

**Figure 7.** Mean biases (solid circles) and standard deviations (open circles) of the differences between OMI and MLS stratospheric ozone columns (SOCs) above 100 hPa (top), 215 hPa (middle) and 261 hPa (bottom) as a function of latitude, during 2004-2008 (black) and 2009-2014 (red).





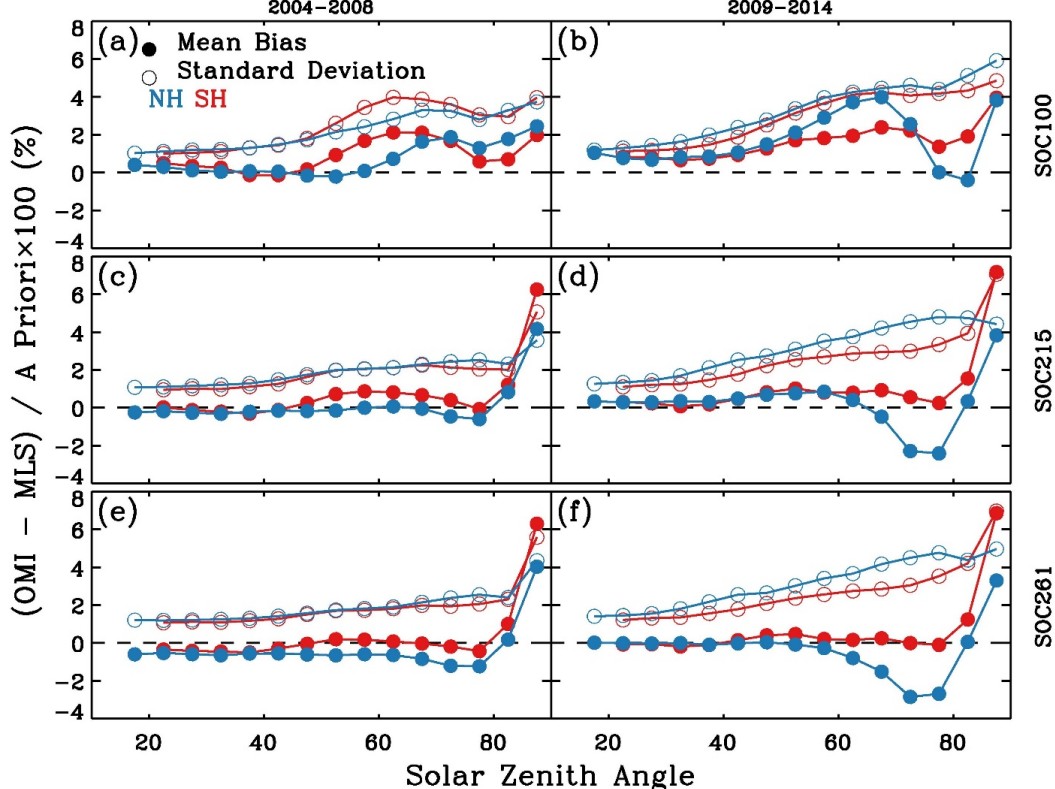

**Figure 8.** Mean biases (solid circles) and standard deviations (open circles) of the differences between OMI and MLS stratospheric ozone columns (SOCs) above 100 hPa (top), 215 hPa (middle) and 261 hPa (bottom) as a function of solar zenith angle for Northern Hemisphere (blue) and Southern Hemisphere (red) during 2004-2008 (left) and 2009-2014 (right).





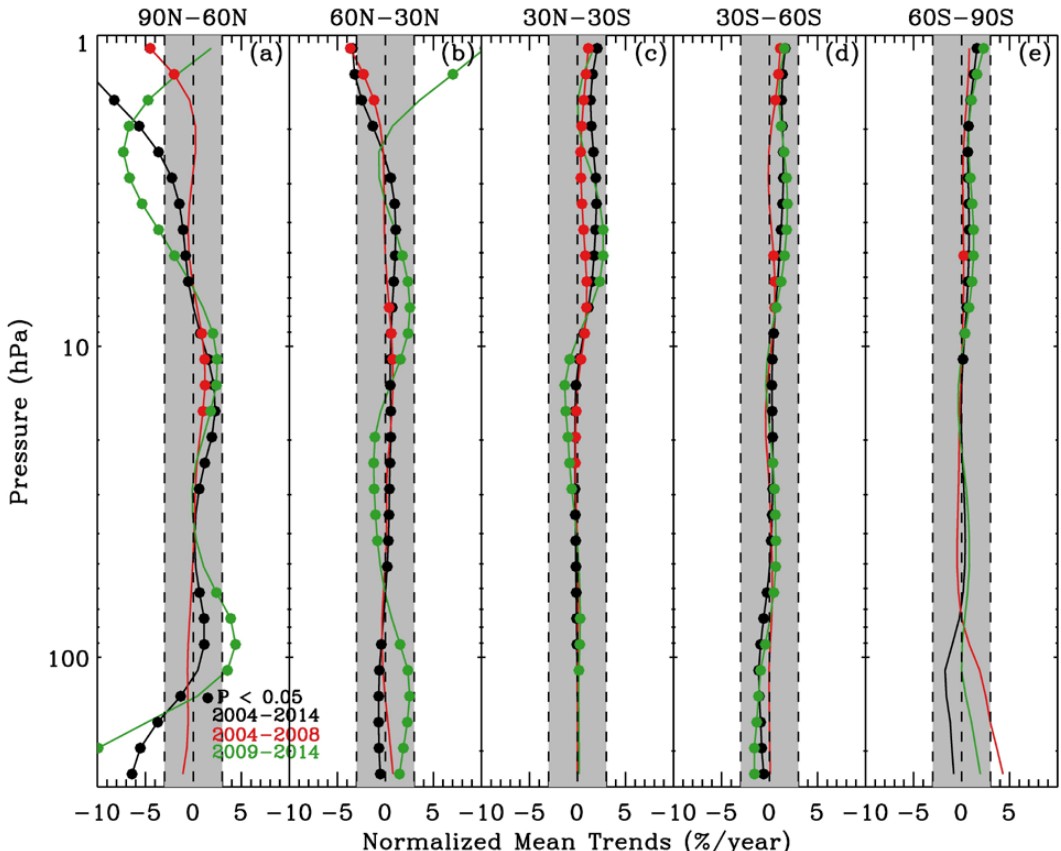

**Figure 9.** Linear regression trends of relative OMI-MLS biases as a function of altitude for five latitudes bands during three time periods: 2004-2014 (black), 2004-2008 (red) and 2009-2014 (green). Solid circles indicate significant trends (with P value less than 0.05).


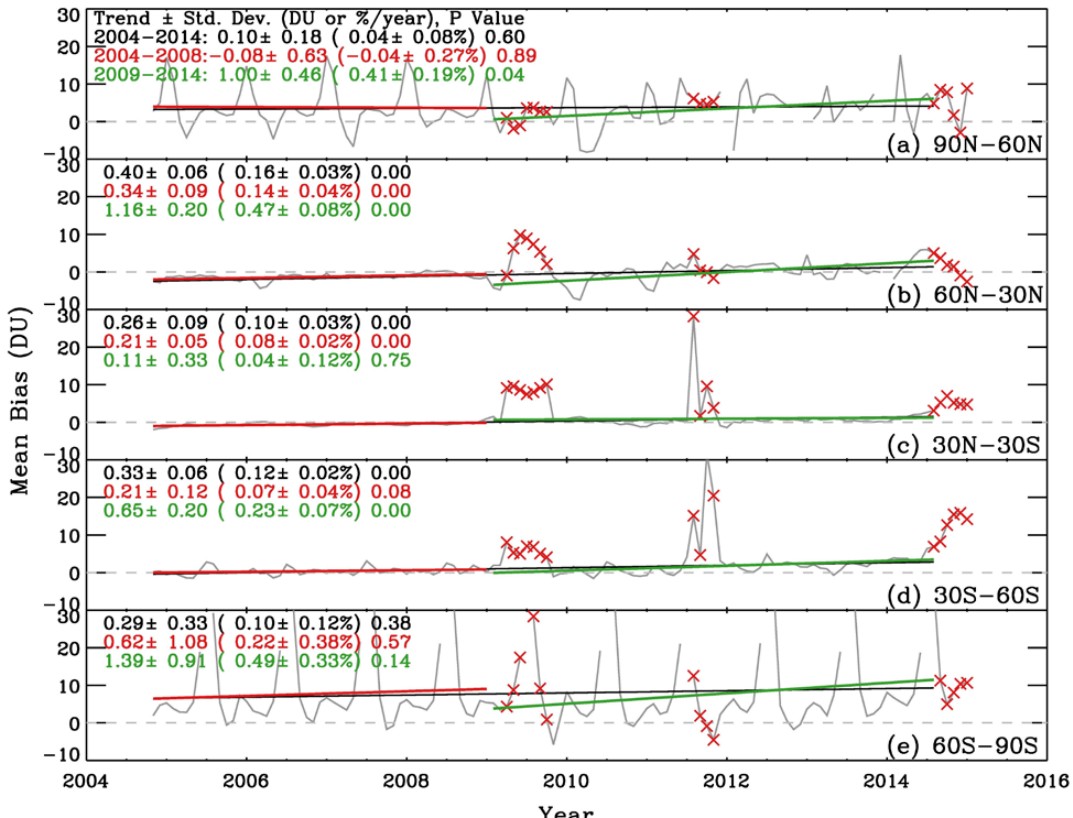

**Figure 10.** Monthly mean biases of OMI-MLS stratospheric ozone columns with pressure less than 215 hPa are plotted in solid gray lines for five different latitude ranges. The corresponding linear trends in 2004-2014, 2004-2008 and 2009-2014 are shown in black, red and green lines, respectively. The red cross signs indicate the periods when OMI has unusually large SOC errors likely due to row anomaly or other unknown issues; the values during these periods are not included in our trend calculations. The legends show linear trends in both DU/year and %/year and the corresponding P values.