# Peer review of "Validation of 10-year SAO OMI Ozone Profile (PROFOZ) Product Using Aura MLS Measurements"

_Atmospheric Measurement Techniques, 2017_

## Referee Comment (RC1) · Anonymous Referee #1 · 6 Jun 2017

I. General impression:

As a follow-up to initial validation work by Liu et al. in 2010, this work reports on the comparative validation with respect to AURA MLS measurements of 10 years of SAO OMI nadir ozone profile data. It thus nicely complements a recent validation exercise of the same OMI data with respect to ozonesonde measurements (Huang et al., 2017). The impact of the occurrence of a serious OMI row anomaly in January 2009 is well addressed, and the comparative analysis is insightfully adopted accordingly. The only major thing missing seems to be a clear motivation for the comparison grid that has been used (see details below). Additionally, it is believed that the clarity of the presentation of the results could be improved by slightly adopting some of the figures, and possibly by including a summary table.

II. Specific comments:

In abstract/introduction and throughout the text, please mention the validated SAO algorithm/product version, as is done for the MLS data, for traceability and for comparison with the results presented in Liu et al. (2010).

Introduction, page 2, lines 35-36: The mentioning of a "suggestion that the possible affection of OMI measurements at shorter wavelengths in the UV-1 channel may have been affected by the RA at all cross-track positions" lacks any notion on how the affection could take place. This seems important however for the succeeding validation motivation. Please provide an appropriate indication of the 'affection' source.

Section 2: The major motivation for the use of MLS data for comparison is its location on the same platform. This instrument however measures microwave thermal emission, whereas OMI is a UV/VIS spectrometer. Please provide a brief sketch of how this wavelength mismatch might affect the comparison results.

Section 3, page 6, lines 1-3: The apparent mismatch between the 0.35 hPa to TOA (line 2) and 0.22 to 261 hPa (line 3) ranges is only explained in lines 23-28 on the same page. Please combine these paragraphs for clarity.

Section 3, page 6, lines 4-22: Although in agreement with Liu et al. (2010), the OMI and MLS profile matching and comparison approaches (still) raise some questions, especially regarding the choice of the MLS grid as the comparison grid: (1) Why opt for the MLS grid as the comparison grid, when OMI is the instrument to be validated? (2) The complementary validation paper using ozonesonde data (Huang et al., 2017) makes use of the OMI grid as the comparison grid, so comparison of results in both papers would be simplified by using the same grid (using the sonde grids would indeed not make sense). (3) Opting for the finer-resolution grid seems to contradict the OMI DFS, which is of the order of 6-7. This means that all grids with a higher resolution than six to seven layers introduce information that is not in the measurement (i.e. coming from the prior). Why then make the grid even finer than the OMI retrieval grid? (4)

[Figure]

Taking the MLS grid as the comparison grid increases the number of operations that has to be performed on the data, hence also the uncertainties: - OMI grid comparison: MLS data to OMI grid, AK-smoothing, Comparison - MLS grid comparison: MLS data to OMI grid, AK-smoothing, MLS data to MLS grid, OMI data to MLS grid, OMI prior to MLS grid, Comparison (5) Returning to the MLS grid after smoothing might indeed make the calculation of relative differences using the MLS data as a reference less stable. The authors' choice for the OMI prior profile as reference in the relative difference denominator is legitimate but somewhat unfortunate. Therefore, (a) please provide a strong motivation for the use of the MLS grid as the comparison grid, taking into account the consequences thereof as described above, and (b) possibly add a plot that clarifies the different vertical grids and comparison ranges (as e.g. in Hubert et al. (2016) Fig. 1). An indication of the approximate OMI unit-DFS layers in this plot would be a very nice surplus.

Section 4.1, Fig. 3(e): Please provide an indication of why the OMI bias exceeds the prior bias within the 1-10 hPa range.

Section 5, page 12, lines 20-21: "The impact of using different coincidence criteria on the ozone profile comparison is negligible" is indeed correct for the different criteria considered in this work, but to what extent (spatio-temporal distance) is this true? Please briefly clarify.

Section 5, page 12, lines 24-26: "The much worse comparison at higher altitudes indicates that RA significantly affects the UV-1 channel of the OMI measurements, although they are not flagged as RA pixels." Based on these results, have you considered introducing an additional L1/L2 flagging/filtering criterion? If so, it would be helpful to mention this.

III. Technical corrections:

Introduction, page 2, line 13: Move link to references and provide access details to capture future changes?

Section 2.1, page 3, lines 31-32: Please use consistent spelling for "cross track-positions" or "cross-track positions" throughout the text.

Section 3, page 6 line 6: Please provide reference to "a procedure provided by the MLS team"

Section 5, page 12, line 30: "straylight" instead of "staylight"

Presentation of results: (1) It might be helpful for the reader to include a comparison result overview table next to the plots. Rows: bias, spread for pre-RA, post-RA for pre-RA period, post-RA Columns: profiles, 100 hPa SOC, 215 hPa SOC, 261 hPa SOC And possibly: Do these numbers comply with the OMI instrument targets? (2) Figure 1: Horizontal lines could indicate the ideal colocation windows and enforced post-RA windows for clarity. By use of interpolated colors, the discreteness of the cross-track position is flawed: E.g. "red" values appear through interpolation, but are not in the data. (3) Figure 2: Vertical lines or gray areas that mark the periods that have been left out of the analysis would be helpful for interpretation of the later plots. (4) Figures 7 and 8: Standard deviations are often plotted with respect to the mean biases, but are plotted with the zero-line as a reference here. Please mention this in the figure captions. (5) Figure 9: The gray areas are not specified in the figure caption. (6) Figure 10: Please in the text or in the figure caption provide an indication of the OOM of the P-values that are now indicated as "0.00".

---

## Referee Comment (RC2) · Anonymous Referee #2 · 10 Aug 2017

This paper is well thought out and contains a thorough analysis of the differences between OMI and MLS profile retrievals. The writing can be somewhat "dense" at times and this reviewer suggests that some of the long, highly complex sentences be split into two to make the reading less difficult. Other than a few minor changes listed below, this manuscript is recommended for publication.

Well, all my underlines and color have disappeared. I hope that you can follow my changes.....

Minor changes: Page 1: line 17-19 remove the words "larger" and "smaller" Larger than what? Line 20,23 & 25 comparisons Line 25 significant bias in the Line 28-9 The sentence about 261hPa MLS ozone sounds very "arrogant" as if MLS is being validated with OMI and not the other way around. Suggestion- just state that they agree well at

this pressure and leave out the interpretation as you have done on Page 5.

Page 2: line 11 change 'in' to 'at' Line 29 ozonesondes measure Line 32 remove "but also. . ..validated"

Page 3 line 22 comparisons

Page 4, line 21 remove "in the stratosphere" Line 21-22: Please either explain how the cross track position changes as a function of latitude here or refer to section 3.

Page 6, line 1 change is to are (data is plural) Line 2 to the top of Line 3 change "for avoiding" to "to avoid" Line 20: OMI a priori is used in the calculation Line 24: The ozone column Line 27: remove "SOC comparison" and add an 's' to "are for comparisons"

Page 7, Line 2,3 Remove "only, missing to. . .. . .than 7hPa" Line 13:mask which is consistant Line 32: (red lines), the OMI

Page 8 line 3: change to: "these comparisons are similar to the OMI/MLS comparisons shown in 2006 in Lui et al although both OMI and MLS versions are now different and this study is done. . .. ." Line 15: change "Such worse comparison" to "The larger differences" Line 24-5: please reference cross track biases or include a plot. Line 33: This supports the theory that. . ..

Page 9: line 11: remove "especially"

Page 10: Line 30: as in the introduction, please reduce the conclusion to "the two agree at 261 level"

Page 11: line 1 add "see Figure 2" after "trend analysis" Line 7 & 8 change "of" to "at" Line 10- you need stronger words to dissuade people from using the data in the upper strat from 30-90N Line 24: . . .not suitable for trend studies. . .. This is a conclusion and should be either moved or repeated in the conclusion section.

Page 12: line 10: profiles

Page 13: remove "original"

Figure 4 & 5: Why is there such a positive bias in 70-90 South above 1 and below 100 hPa plot in figure 5 (SZA) but not figure 4?? Shouldn't it "smear out" and be a red streak in the latitude plot like in the north high latitudes (Fig 4)? Please explain.

Figure 6 is very "cluttered" with text. Please remove N= for the lower two plots as it is redundant information.

Figure 7: Please scale the middle plot to the same absolute scale as the other two (-4 to 6 or -3 to 7 would be fine)

---

## Author Response (AR1)

Responses to Referee #1:

We thank referee's helpful and constructive comments and review. The referee's comments are listed in *italics*, and our responses in black with revised texts in **bold black**. Please noted that figure numbers are different from those in the original manuscripts.

*I. General impression:*

*As a follow-up to initial validation work by Liu et al. in 2010, this work reports on the comparative validation with respect to AURA MLS measurements of 10 years of SAO OMI nadir ozone profile data. It thus nicely complements a recent validation exercise of the same OMI data with respect to ozonesonde measurements (Huang et al., 2017). The impact of the occurrence of a serious OMI row anomaly in January 2009 is well addressed, and the comparative analysis is insightfully adopted accordingly. The only major thing missing seems to be a clear motivation for the comparison grid that has been used (see details below). Additionally, it is believed that the clarity of the presentation of the results could be improved by slightly adopting some of the figures, and possibly by including a summary table.*

*II. Specific comments:*

*In abstract/introduction and throughout the text, please mention the validated SAO algorithm/product version, as is done for the MLS data, for traceability and for comparison with the results presented in Liu et al. (2010).*

We have added the version number (v0.93) for the current product. Note that the product of Liu et al. (2010) does not have a version as it is a research product that is not produced routinely, but with very limited spatiotemporal coverage.

*Introduction, page 2, lines 35-36: The mentioning of a "suggestion that the possible affection of OMI measurements at shorter wavelengths in the UV-1 channel may have been affected by the RA at all cross-track positions" lacks any notion on how the affection could take place. This seems important however for the succeeding validation motivation. Please provide an appropriate indication of the 'affection' source.*

We have added some notion and a reference as follows:

"…been affected **by blockage and solar radiation effects** of the RA… **(Schenkeveld et al., 2017).**"

*Section 2: The major motivation for the use of MLS data for comparison is its location on the same platform. This instrument however measures microwave thermal emission, whereas OMI is*

*a UV/VIS spectrometer. Please provide a brief sketch of how this wavelength mismatch might affect the comparison results.*

OMI and MLS have very different vertical sensitivities on the ozone profiles due to the different wavelengths of these sensors deployed for retrievals and mostly due to different observation modes (nadir for OMI vs. limb for MLS). These observation differences result in the different vertical resolutions and grids. Therefore, we also include the comparison between OMI and MLS with applying OMI averaging kernels due to much higher vertical resolution in the MLS data to account for differences in vertical sensitivities.

We have added **"…due to the use of observation modes (nadir for OMI and limb for MLS) and different spectral regions …" in P6 L4 of previous manuscripts.**

*Section 3, page 6, lines 1-3: The apparent mismatch between the 0.35 hPa to TOA (line 2) and 0.22 to 261 hPa (line 3) ranges is only explained in lines 23-28 on the same page. Please combine these paragraphs for clarity.*

We have combined these paragraphs.

*Section 3, page 6, lines 4-22: Although in agreement with Liu et al. (2010), the OMI and MLS profile matching and comparison approaches (still) raise some questions, especially regarding the choice of the MLS grid as the comparison grid: (1) Why opt for the MLS grid as the comparison grid, when OMI is the instrument to be validated? (2) The complementary validation paper using ozonesonde data (Huang et al., 2017) makes use of the OMI grid as the comparison grid, so comparison of results in both papers would be simplified by using the same grid (using the sonde grids would indeed not make sense). (3) Opting for the finer-resolution grid seems to contradict the OMI DFS, which is of the order of 6-7. This means that all grids with a higher resolution than six to seven layers introduce information that is not in the measurement (i.e. coming from the prior). Why then make the grid even finer than the OMI retrieval grid? (4) Taking the MLS grid as the comparison grid increases the number of operations that has to be performed on the data, hence also the uncertainties: - OMI grid comparison: MLS data to OMI grid, AK-smoothing, Comparison - MLS grid comparison: MLS data to OMI grid, AK-smoothing, MLS data to MLS grid, OMI data to MLS grid, OMI prior to MLS grid, Comparison (5) Returning to the MLS grid after smoothing might indeed make the calculation of relative differences using the MLS data as a reference less stable. The authors' choice for the OMI prior profile as reference in the relative difference denominator is legitimate but somewhat unfortunate. Therefore, (a) please provide a strong motivation for the use of the MLS grid as the comparison grid, taking into account the consequences thereof as described above, and (b) possibly add a plot that clarifies the different vertical grids and comparison ranges (as e.g. in Hubert et al. (2016) Fig. 1). An indication of the approximate OMI unit-DFS layers in this plot would be a very nice surplus.*

(a)

We have tested the validation by using OMI grid as shown in the figure below. As has been expected, the mean biases and standard deviations with OMI grids are slightly better than those with MLS grids, but both results with OMI and MLS grids have very similar morphology. The clearly larger values with the MLS grid near the top and bottom are due to smaller MLS layers and broader rangers at both ends than cannot be compared with the use of OMI grid. There are a number of reasons to adopt the use of MLS grid in Liu et al. (2010) in addition to that retrievals can be validated at slightly broader altitude ranges.

(1) We want to show that OMI retrievals, despite coarser vertical resolutions (but smaller retrieval errors), compare quite well with MLS data that have much better vertical resolution) even at MLS vertical grid and without applying OMI averaging kernels. This is not to show that OMI have enough vertical resolution comparable to MLS, but show that OMI retrievals, constrained by climatological a priori, agree quite well with MLS data at MLS grid. It should be noted that the a priori already agrees with MLS data to the 10% level in the middle stratosphere. Our retrieval uses this tight a priori constrain to minimize retrieval error although it decreases the DFS/resolution compared to use of a larger a priori constraint.

(2) Using the MLS grid facilitates the comparison of stratospheric ozone columns down to MLS pressure levels like 100 hPa or 215 hPa as MLS stratospheric ozone columns have been used to derive tropospheric ozone column in combination with OMI total ozone. Using the OMI grid, we cannot directly integrate the ozone down to 100 hPa and 215 hPa from the convolved MLS profiles unless we interpolate it back to the MLS grid or use some portion of the original MLS profile that cannot be convolved due to OMI/MLS altitude grid mismatch.

So in this study, we continue the use of OMI grid for contrast with the results in Liu et al. (2010). Also, we would like to cross-validate MLS stratospheric ozone columns down to 261 hPa and MLS ozone from 215 hPa to 261 hPa because the MLS official documents suggest that MLS ozone at 261 hPa still requires further evaluation for scientific use. Using the MLS grid will facilitate these additional comparisons as well.

We added the following sentence to P6 L5 of the original manuscript: "**In contrast to typical profile comparison using the vertical grid of the product with coarser vertical resolution, we conduct the comparison using the finer MLS vertical grid following Liu et al. (2010a) to demonstrate that OMI retrievals compare quite well with MLS data even at the MLS grid and facilitate the cross-validation of ozone from 215 hPa to 261 hPa and stratospheric ozone columns integrated down to several MLS pressure levels.**"

[Figure]

**Figure: Similar to Figure 4, but with OMI grids.**

(b) We have added Figure 2 to clarify the vertical grids and comparison ranges. Because this paper focuses on the validation, we have added a reference Liu et al. (2010) for readers who are interested in DFL profiles. We have revised it as "More details on the PROFOZ product and ozone profile retrieval algorithm, **including the unit Degree of Freedom for Signal (DFS) profiles,** can be found in Huang et al. (2017) and Liu et al. (2010)."

[Figure]

**"Figure 2. Overview of the vertical range of OMI and MLS. The solid black bars represent vertical grids of OMI and MLS, respectively, while the validation range of this study is marked by grey lines."**

*Section 4.1, Fig. 3(e): Please provide an indication of why the OMI bias exceeds the prior bias within the 1-10 hPa range.*

This is due to the same reason mentioned shortly afterwards that RA affects radiance in the UV-1 channel. We have added as follows "**In addition, the retrieval biases are larger those in the a priori biases within 1-10 hPa**" before explaining the degradation of the results.

*Section 5, page 12, lines 20-21: "The impact of using different coincidence criteria on the ozone profile comparison is negligible" is indeed correct for the different criteria considered in this work, but to what extent (spatio-temporal distance) is this true? Please briefly clarify.*

We didn't further test to find out the spatiotemporal extent to which different coincidence criteria can be considered. As we discussed in the first paragraph of Sect. 3, the OMI cross-track position collocated with MLS data varies from position 20 in the tropics to position 15 at high latitudes. After the occurrence of RA, the cross-track positions vary from position 22 in the tropics to position 12 at high latitudes. For most cases, the distance shift between two different coincident criteria is approximately 100-200 km; for some extreme cases, the maximum distance shift is ~300-400 km. The variability of stratospheric ozone is relatively small on average within these distances. However, if the RA affected pixels spread in the future, this assumption would not be necessarily true. Ozone variability and longer distance will increase the standard deviations of the differences. In addition, there are cross-track dependent biases in our product especially for those extreme off-nadir positions as shown in Huang et al. (2017); collocating OMI retrievals at these extreme off-nadir positions with MLS data will further increase the standard deviations of the OMI/MLS differences.

*Section 5, page 12, lines 24-26: "The much worse comparison at higher altitudes indicates that RA significantly affects the UV-1 channel of the OMI measurements, although they are not flagged as RA pixels." Based on these results, have you considered introducing an additional L1/L2 flagging/filtering criterion? If so, it would be helpful to mention this.*

This validation study will caution the users about using the ozone profile retrievals at high altitudes during the post-RA period. We will suggest to the instrument calibration team to see if they can add this additional flagging in the level 1b data based on instrument calibration. However, we are not considering it by ourselves because of the following reasons. A good flagging criterion should be preferably based on radiances in the level 1b data or instrument calibration process. The UV1 channel is mainly used for ozone profile retrievals, not used in most of the OMI products. In addition, we did not validate retrievals all cross-track positions except for those positions collocated with MLS data. Furthermore, despite of larger biases and standard deviations at higher altitudes, retrievals for pressure > 10 hPa and stratospheric ozone columns are not significantly degraded and can still be useful.

*III. Technical corrections:*

*Introduction, page 2, line 13: Move link to references and provide access details to capture future changes?*

Done.

*Section 2.1, page 3, lines 31-32: Please use consistent spelling for "cross track positions" or "cross-track positions" throughout the text.*

Done. We use "cross-track positions" in this paper.

*Section 3, page 6 line 6: Please provide reference to "a procedure provided by the MLS team"*

Done. We added Liu et al. (2010a).

*Section 5, page 12, line 30: "straylight" instead of "staylight"*

Done.

*Presentation of results: (1) It might be helpful for the reader to include a comparison result overview table next to the plots. Rows: bias, spread for pre-RA, post-RA for preRA period, post-RA Columns: profiles, 100 hPa SOC, 215 hPa SOC, 261 hPa SOC And possibly: Do these numbers comply with the OMI instrument targets?*

The comparison results are all included in Figure 4 and Figure 6, the information would be repeated if we had a table. Generally, all our results comply with the OMI instrument targets. We have added in Sect. 5 as
**"From the OMI scientific requirement document (Levelt et al., 2000), the scientific requirement for OMI ozone is to achieve 10% accuracy in the stratosphere and 30% in the troposphere. The validation of stratospheric ozone profiles with MLS data indicates that our retrieval performance generally meets OMI's scientific requirement in most of the stratosphere especially during the pre-RA period (Levelt, 2000) except for high latitudes/SZAs."** in the 3rd paragraph of Sect. 5.

and

"**Overall, our validation indicates that our retrieval performance in the stratosphere generally meets OMI's scientific requirement for the ozone profile product to have 10% accuracy in the stratosphere (Levelt, 2000).**" in the 5th paragraph of Sect. 5.

*(2) Figure 1: Horizontal lines could indicate the ideal colocation windows and enforced post-RA windows for clarity. By use of interpolated colors, the discreteness of the cross-track position is flawed: E.g. "red" values appear through interpolation, but are not in the data.*

We added horizontal lines and discrete colors as follows:

[Figure]

We have also added in the caption as "… **The solid lines represent the range of OMI-MLS collocations that vary from position 20 in the tropics to 15 at high latitudes, while the dashed lines represent the enforced post-RA OMI-MLS coincidence criteria (the closest OMI pixel from position ≤ 12 or ≥ 22)**"

*(3) Figure 2: Vertical lines or gray areas that mark the periods that have been left out of the analysis would be helpful for interpretation of the later plots.*

We have added gray areas to mark the periods that have been left out as follows:

[Figure]

*(4) Figures 7 and 8: Standard deviations are often plotted with respect to the mean biases, but are plotted with the zero-line as a reference here. Please mention this in the figure captions.*

We have added this in the captions of Figure 8 and 9 as:

**"… Note that standard deviations are plotted with the zero-line as a reference."**

*(5) Figure 9: The gray areas are not specified in the figure caption.*

We have added **"… The grey areas indicate the range of -3% and 3% in each panel."**

*(6) Figure 10: Please in the text or in the figure caption provide an indication of the OOM of the P-values that are now indicated as "0.00".*

Done.

Huang, G., Liu, X., Chance, K., Yang, K., Bhartia, P. K., Cai, Z., Allaart, M., Ancellet, G., Calpini, B., Coetzee, G. J. R., Cuevas-Agulló, E., Cupeiro, M., De Backer, H., Dubey, M. K., Fuelberg, H. E., Fujiwara, M., Godin-Beekmann, S., Hall, T. J., Johnson, B., Joseph, E., Kivi, R., Kois, B., Komala, N., König-Langlo, G., Laneve, G., Leblanc, T., Marchand, M., Minschwaner, K. R., Morris, G., Newchurch, M. J., Ogino, S. Y., Ohkawara, N., Piters, A. J. M., Posny, F., Querel, R., Scheele, R., Schmidlin, F. J., Schnell, R. C., Schrems, O., Selkirk, H., Shiotani, M., Skrivánková, P., Stübi, R., Taha, G., Tarasick, D. W., Thompson, A. M., Thouret, V., Tully, M. B., Van Malderen, R., Vömel, H., von der Gathen, P., Witte, J. C., and Yela, M.: Validation of 10-year SAO OMI Ozone Profile (PROFOZ) product using ozonesonde observations, Atmos. Meas. Tech., 10, 2455-2475, doi: 10.5194/amt-10-2455-2017, 2017.

Levelt, P. F.: Science requirements document for OMI-EOS, KNMI, 2000.

Liu, X., Bhartia, P. K., Chance, K., Spurr, R. J. D., and Kurosu, T. P.: Ozone profile retrievals from the Ozone Monitoring Instrument, Atmos. Chem. Phys., 10, 2521-2537, doi: 10.5194/acp-10-2521-2010, 2010.

Responses to Referee #2:

We thank referee's helpful and constructive comments and review. The referee's comments are listed in *italics*, and our responses in black with revised texts in **bold black**. Please note that figure number has been revised according to Referee #1's comments and suggestions.

*This paper is well thought out and contains a thorough analysis of the differences between OMI and MLS profile retrievals. The writing can be somewhat "dense" at times and this reviewer suggests that some of the long, highly complex sentences be split into two to make the reading less difficult. Other than a few minor changes listed below, this manuscript is recommended for publication.*

*Well, all my underlines and color have disappeared. I hope that you can follow my changes.....*

*Minor changes: Page 1: line 17-19 remove the words "larger" and "smaller" Larger than what?*

This sentence shows the comparison results of the post-RA period with relative to those during the pre-RA period as we show the comparison results during the pre-RA period in the previous two sentences. To avoid confusion, we have changed the sentence to "**Compared to the retrievals during the pre-RA period, OMI retrievals during the post-RA period** …"

*Line 20,23 & 25 comparisons.*

Done.

*Line 25 significant bias*

Done.

*in the Line 28-9 The sentence about 261hPa MLS ozone sounds very "arrogant" as if MLS is being validated with OMI and not the other way around. Suggestion- just state that they agree well at this pressure and leave out the interpretation as you have done on Page 5.*

We deleted this sentence, because we already have the agreement statement in the abstract.

*Page 2: line 11 change 'in' to 'at' Line 29 ozonesondes measure Line 32 remove "but also....validated"*

Done.

*Page 3 line 22 comparisons*

Done.

*Page 4, line 21 remove "in the stratosphere".*

Done.

*Line 21-22: Please either explain how the cross track position changes as a function of latitude here or refer to section 3.*

We have added "The **threshold of** cross-track position **…(More details will be discussed in Sect. 3).**"

*Page 6, line 1 change is to are (data is plural) Line 2 to the top of Line 3 change "for avoiding" to "to avoid" Line 20: OMI a priori is used in the calculation Line 24: The ozone column Line 27: remove "SOC comparison" and add an 's' to "are for comparisons"*

Done.

*Page 7, Line 2,3 Remove "only, missing to......than 7hPa"*

Done.

*Line 13:mask which is consistent*

Done.

*Line 32: (red lines), the OMI*

Done.

*Page 8 line 3: change to: "these comparisons are similar to the OMI/MLS comparisons shown in 2006 in Lui et al although both OMI and MLS versions are now different and this study is done....."*

Done.

*Line 15: change "Such worse comparison" to "The larger differences"*

Done.

*Line 24-5: please reference cross track biases or include a plot.*

We have added (Huang et al., 2017) as a reference.

*Line 33: This supports the theory that....*

Done.

*Page 9: line 11: remove "especially"*

Done.

*Page 10: Line 30: as in the introduction, please reduce the conclusion to "the two agree at 261 level"*

We have removed the sentence and added to the previous sentence **"indicating the scientific use of MLS ozone at 261 hPa in applications such as the OMI/MLS TOR method"**

*Page 11: line 1 add "see Figure 2" after "trend analysis" Line 7 & 8 change "of" to "at"*

Done.

*Line 10- you need stronger words to dissuade people from using the data in the upper strat from 30-90N*

We have revised it as "**… where there are significantly large trends of up to -5%/year…**"

*Line 24: ...not suitable for trend studies.... This is a conclusion and should be either moved or repeated in the conclusion section.*

We have rephrased it in the conclusion section as **"These significant bias trends indicate that the current ozone profile product is not suitable for trend studeis, especially during the post-RA period."**

*Page 12: line 10: profiles*

Done.

*Page 13: remove "original"*

Done.

*Figure 4 & 5: Why is there such a positive bias in 70-90 South above 1 and below 100 hPa plot in figure 5 (SZA) but not figure 4?? Shouldn't it "smear out" and be a red streak in the latitude plot like in the north high latitudes (Fig 4)? Please explain.*

The large positive mean biases with 85°-90° solar zenith angles in the southern hemisphere are due to their large solar zenith angles. In Figure 4, mean biases at southern high latitude regions include not only large mean biases due to the large solar zenith angles during southern winter, but also smaller and/or negative mean biases with smaller solar zenith angles during southern summer. Naturally, large positive biases with large solar zenith angles during southern hemisphere winter are smeared out by more frequent smaller or negative biases with smaller solar zenith angles during other seasons. Also the figure shows very similar patterns between south and north high latitudes.

*Figure 6 is very "cluttered" with text. Please remove N= for the lower two plots as it is redundant information.*

Done.

[Figure]

*Figure 7: Please scale the middle plot to the same absolute scale as the other two (-4 to 6 or -3 to 7 would be fine)*

Done.

[revised manuscript text omitted]

---

## Author Response (AR2)

Responses to Referees:

We thank referees' helpful and constructive comments and review. The referees' comments are listed in *italics*, and our responses in black with revised texts in **bold black**. Please noted that figure numbers are different from those in the original manuscripts.

*The authors have made all the corrections necessary to have this work published. One final suggestion is to clean up the change in "voice" that occurs throughout the paper. There are many sentences using "we" and "our" and then immediately followed by "this" and "the". Eg: "We evaluate the effects…" and then "The retrieval comparisons indicate…." This should be either "This study evaluated the effects … or "Our retrieval comparisons indicate…." Try to keep the same narrative voice. Again, this is just a suggestion to make the paper read better but not absolutely necessary for final publication.*

We have revised them accordingly.

*Specific changes to be made:*

*Page 6 line 8: capitalize 'we'*

Done.

*Page 8 line 4 & 5 You emphasize "red lines" twice. Once is sufficient.*

We deleted the second "(red lines)".

*Figure 4: Are the dashed lines in c) and d) really necessary? They clutter up the plot in my opinion.*

These dashed lines are necessary. They indicate negligible differences on mean bias profiles between with and without applying pre-RA masks. This is an important assumption in this paper. We keep these dashed lines in Figure 4 (c) and (d).

[revised manuscript text omitted]